# Apoptosis of Pancreatic Cancer Cells after Co-Treatment with Eugenol and Tumor Necrosis Factor-Related Apoptosis-Inducing Ligand

**DOI:** 10.3390/cancers16173092

**Published:** 2024-09-05

**Authors:** Hyun Hee Kim, Suk-Young Lee, Dae-Hee Lee

**Affiliations:** 1Department of Marine Bio Food Science, Gangneung-Wonju National University, Jukheon-gil 7, Gangneung 25457, Republic of Korea; myeoni@gwnu.ac.kr; 2Department of Pathology, Korea University Guro Hospital, 148, Gurodong-ro, Guro-gu, Seoul 08308, Republic of Korea

**Keywords:** TNF-related apoptosis-inducing ligand, eugenol, death receptor 5, endoplasmic reticulum stress, ROS, siRNA

## Abstract

**Simple Summary:**

Pancreatic cancer is a challenging disease with limited treatment options and poor prognosis. This study aims to investigate whether eugenol, the main component of clove oil, can enhance the effectiveness of tumor necrosis factor-related apoptosis-inducing ligand (TRAIL). Eugenol is known for its anticancer properties, and we aim to determine if it can increase the expression of death receptors to reduce TRAIL resistance. By exploring the combination of eugenol and TRAIL, we seek to understand how effective this approach is in inducing apoptosis in pancreatic cancer cells. The findings could offer a new strategy for treating pancreatic cancer and provide significant insights into therapeutic approaches for this aggressive disease within the research community.

**Abstract:**

Pancreatic cancer is a refractory cancer with limited treatment options. Various cancer types are resistant to tumor necrosis factor-related apoptosis-inducing ligand (TRAIL). Eugenol, the main component of clove oil, exhibits anticancer, anti-inflammatory, and antioxidant effects. However, no studies have reported that eugenol increases TRAIL sensitivity by upregulating death receptor (DR) expression. Here, we aimed to investigate eugenol as a potent TRAIL sensitizer. Increased apoptosis and inhibition of cell proliferation was observed in pancreatic cancer cells treated with eugenol and TRAIL compared with those treated with eugenol alone. Eugenol upregulated the expression of DR5, inhibited the FLICE-inhibitory protein (FLIP), an anti-apoptotic protein, and increased p53, a tumor suppressor protein. In addition, eugenol induced the generation of reactive oxygen species (ROS) and caused endoplasmic reticulum (ER) stress. C/EBP-homologous protein (CHOP) knockdown using siRNA decreased the expression of DR5 and reduced the combined effects of eugenol and TRAIL. These results demonstrate that eugenol enhances TRAIL-induced apoptosis by upregulating DR5 through the ROS-mediated ER stress–CHOP pathway, which enhances ER stress by inducing p53 and downregulating FLIP expression. This suggests that eugenol has the potential to treat pancreatic cancer by increasing cell sensitivity to TRAIL.

## 1. Introduction

Pancreatic cancer has a low global incidence; however, its survival rate is also low, as many cases are already advanced when discovered. In addition, pancreatic cancer tends to spread quickly into surrounding tissues and infiltrate the surrounding blood vessels or adjacent organs, limiting the opportunities for surgical treatment. Existing anticancer drugs for pancreatic cancer have severe side effects and do not significantly increase the survival time of stage-4 patients [1]. Although numerous chemotherapeutic regimens have been used clinically to treat pancreatic cancer, successful treatment necessitates the development of novel therapeutic agents.

Tumor necrosis factor (TNF)-related apoptosis-inducing ligand (TRAIL) is a cytokine produced by various immune cells. TRAIL induces cell death by interacting with the death receptors TRAIL-R1/DR4 and TRAIL-R2/DR5, which are overexpressed in cancer cells [2,3]. TRAIL is a tumor-selective agent that activates the signaling pathway used by the innate immune system to bind to death receptors on cancer cells and induce apoptosis [3,4]. It is non-toxic to normal cells, primarily due to their overexpression of decoy receptors [5,6,7]. Therefore, TRAIL has therapeutic potential for cancer treatment and is currently undergoing phase II clinical trials [8,9,10]. However, various cancer cells, including pancreatic cancer cells, are resistant to TRAIL. The main causes of TRAIL resistance are the inhibition of DR4 and DR5 due to the overexpression of DcR1/2 in cancer cell lines and anti-apoptotic proteins, which inhibit the action of pro-apoptotic proteins in cells [11,12]. Therefore, it is essential to understand the fundamental mechanisms underlying TRAIL resistance and how to overcome this resistance. TRAIL can be used in combination with other anticancer drugs or natural products to increase the TRAIL sensitivity of cancer cells [3,13].

FLICE-inhibitory protein (FLIP) is an anti-apoptotic protein that has two different forms, FLIP(L) and FLIP(S), which both inhibit cell death by interacting with caspase-8 [14,15]. The tumor suppressor protein p53 induces the accumulation of ROS in mitochondria and increases intracellular ROS levels, thereby increasing C/EBP-homologous protein (CHOP) expression [16,17]. Excessive production of reactive oxygen species (ROS) within the endoplasmic reticulum (ER) and accumulation of misfolded proteins in the ER lumen lead to ER stress. This stress serves as a defense mechanism for cell survival [3]. However, severe or prolonged ER stress can trigger apoptosis [18]. Multiple studies have demonstrated that CHOP is one of the most highly induced genes during ER stress. It also regulates the expression of DR5 [5,19,20].

Eugenol, the main component of clove oil, is extracted by steam distillation from the dried flower buds of clove trees and has been used as a spice, painkiller, and disinfectant. According to a recent study, eugenol exhibits anticancer, anti-inflammatory, and antioxidant effects [21]. Its anticancer effects in colon and lung cancers have also been reported [22,23]. However, no studies have investigated whether eugenol increases TRAIL sensitivity by changing DR expression. 

Therefore, this study aimed to determine whether eugenol increases DR expression in pancreatic cancer cells and whether the combined treatment with eugenol and TRAIL increases apoptosis in cancer cells compared to treatment with either eugenol or TRAIL. Our findings showed that eugenol enhanced TRAIL-induced apoptosis by upregulating DR5 via the ROS-mediated ER stress–CHOP pathway. In addition, we demonstrated that eugenol induced the tumor suppressor protein p53, enhanced ER stress, and downregulated the expression of FLIP. Our findings indicate that eugenol is a new sensitizer that enhances sensitivity to TRAIL.

## 2. Materials and Methods

### 2.1. Cell Culture

The human pancreatic cancer cell lines used in this study were PANC-1, BxPC-3, Panc 02.03, MIA-PaCa-2, Panc 10.05, and AsPC-1, which were provided by Dr. Kim (Asan Medical Center, Seoul, Republic of Korea). Cells were cultured as monolayers in Dulbecco’s modified Eagle’s Medium (DMEM; Gendepot, Katy, TX, USA), which was supplemented with 10% fetal bovine serum (Gibco, Miami, FL, USA), 1 mM L-glutamine, and 1% penicillin/streptomycin (Lonza, Basel, Switzerland). Normal human pancreatic cells (HPDE) and breast cancer (MDA-MB-231, MCF-7), lung cancer (LN-229, U-251), and colorectal cancer (HT-29, HCT116) cell lines were purchased from ATCC. All cell lines were cultured in an incubator with a temperature of 37 °C and 5% CO_2_.

### 2.2. Reagent and Antibodies

Eugenol was purchased from Sigma-Aldrich (St. Louis, MO, USA), dissolved in DMSO, and stored at –20 °C. Recombinant Human sTRAIL/Apo2L was purchased from PeproTech (Rocky Hill, NJ, USA). The drug was dissolved in PBS and stored at −80 °C. Anti-PARP, anti-cleaved caspase-8, anti-cleaved caspase-3, anti-HSP90, anti-FLIP, anti-XIAP, anti-Mcl-1, anti-Bcl-XL, anti-Bcl-2, anti-Bax, anti-Survivin, anti-DR5, anti-DR4, anti-PERK, anti-ATF-6, anti-BiP, anti-p53, anti-ATF-4, anti-eIF2α, anti-p-eIF2α, anti-CHOP, and anti-β-actin antibodies were purchased from Cell Signaling Technology (Danvers, MA, USA). The anti-p-PERK antibody was purchased from Santa Cruz Biotechnology (Santa Cruz, CA, USA). The anti-p-IRE-1α antibody was purchased from Novus Biologicals (Littleton, CO, USA). Secondary anti-mouse IgG horseradish peroxidase (HRP) and anti-rabbit IgG HRP antibodies were purchased from Cell Signaling Technology (Danvers, MA, USA). The EZ-Cytox cell viability assay kit was purchased from DoGen Bio (Seoul, Republic of Korea), and the Annexin V-FITC apoptosis detection kit was purchased from Koma Biotech (Seoul, Republic of Korea). Paraformaldehyde (4%) was purchased from Tissue-Pro Technologies. The crystal violet solution was purchased from Sigma-Aldrich (St. Louis, MO, USA). The caspase-Glo^®^ 3/7 Assay System was purchased from Promega (Madison, WI, USA). The dichloro-dihydro-fluorescein diacetate (DCFH-DA) was purchased from Sigma-Aldrich (St. Louis, MO, USA).

### 2.3. WST-8 Assay

To determine cell proliferation, cells were seeded in 96-well plates at a density of 1 × 10^4^ cells/well and then cultured at 37 °C. Cells were treated with either 400 µM eugenol, 100 ng/mL TRAIL, or both combined. Cell proliferation was analyzed using an EZ-Cytox assay (DoGen Bio, Seoul, Republic of Korea). Ten microliters of EZ-Cytox solution were added to each well, and the plates were incubated for 2 h. Absorbance was measured at 450 nm using a microplate reader (Beckman Coulter, Brea, CA, USA).

### 2.4. Crystal Violet Assay

The cells were seeded in 6-well plates at a density of 1.5 × 10^5^ cells/well and incubated in a 37 °C incubator for 24 h. Cells were treated with either 400 µM eugenol, 100 ng/mL TRAIL, or both combined. The cells were washed with PBS and fixed with 4% paraformaldehyde for 15 min; then, the colonies were fixed with ice-cold methanol for 15 min. The cells were stained with crystal violet for 15 min for visualization. After staining, the cells were allowed to dry, and methanol was subsequently added to dissolve the dye. The absorbance was measured at 590 nm.

### 2.5. Colony Formation Assay

The cells were seeded at a density of 500 cells/well in 6-well plates and cultured at 37 °C. Cells were treated with either 400 µM eugenol, 100 ng/mL TRAIL, or both combined. The medium was changed every 3 days. After 2 weeks, the cells were washed with PBS. The cells were then fixed with 4% paraformaldehyde for 15 min; then, the colonies were fixed for 15 min with ice-cold methanol. Finally, the cells were stained with crystal violet for 15 min for visualization and counting.

### 2.6. Wound Healing Assay

Cells were cultured in 96-well plates at 3 × 10^4^ cells/well to confluent monolayers overnight. Straight wounds were created using Wound Maker™ (Essen BioScience, Ann Arbor, MI, USA). After washing with PBS to remove cell debris and treatment with 400 μM eugenol and 100 ng/mL TRAIL, the wound gaps were photographed every 2 h between 0 and 24 h, and the area of cell-free wounds was measured using the IncuCyte^®^ Live-Cell Analysis System (Essen BioScience, Ann Arbor, MI, USA).

### 2.7. Flow Cytometry Analysis

The movement of phosphatidylserine, a marker of apoptosis, from the inner to outer layer of the plasma membrane was identified through the binding of Annexin V conjugate with fluorescein isothiocyanate (FITC). PANC-1 cells were seeded at a density of 1.5 × 10^5^ cells/well in 6-well plates and cultured overnight. The cells were then treated with either 400 μM eugenol, 100 ng/mL TRAIL, or both combined. The apoptosis assay was performed according to the manufacturer’s instructions. The cells were harvested by centrifugation at 1000× *g* for 5 min after treatment with trypsin-EDTA. The supernatant was removed, washed with ice-cold PBS, and centrifuged again at 1000× *g* for 5 min. After centrifugation, 500 μL of 1× binding buffer was added to resuspend the cells. Next, the cells were incubated with Annexin V (1.25 μL) at room temperature (22 °C) in a dark room for 15 min. After 15 min, the mixture was centrifuged at 1000× *g* for 5 min. The supernatant was then removed, and 500 μL of 1× binding buffer and 10 μL of propidium iodide (PI) were added. Apoptosis was immediately measured using a CytoFLEX flow cytometer (Beckman Coulter, Brea, CA, USA). Additionally, to analyze the expression of DR4 and DR5 on the cell surface, live cells were cultured with each antibody and analyzed using flow cytometry.

### 2.8. Western Blotting

PANC-1 cells were seeded in 60 mm plates at 4.5 × 10^5^ cells/well, cultured overnight, and then treated with eugenol and TRAIL for 24 h according to the determined concentration. The cells were lysed with 100 μL of 2× SDS-sample buffer, and protein concentrations were measured using the bicinchoninic acid (BCA) assay. Equal amounts of protein were separated by SDS-PAGE and transferred to PVDF membranes. The membranes were blocked with TBS containing 0.2% Tween-20 and 5% skim milk. The primary antibody was incubated at 4 °C overnight. After washing, the samples were incubated with a secondary antibody conjugated to HRP for 1 h. Protein expression was visualized with an ECL solution using Amersham ImageQuant 800 (Cytiva, Uppsala, Sweden).

### 2.9. Caspase 3/7 Assay

PANC-1 cells were seeded at a density of 1 × 10^4^ cells/well in 96-well plates and cultured at 37 °C. Cells were treated with either 400 µM eugenol, 100 ng/mL TRAIL, or both combined. Caspase-Glo^®^ 3/7 reagent (100 μL) was added to each well of a 96-well plate with white walls. The plate was gently shaken at 300–500 rpm for 30 s to mix the contents. Then, the cells were incubated at room temperature (22 °C) for 1 h. The luminescence of each sample was measured using a Synergy HTX Multi-Mode Reader (BioTek Instruments, Winooski, VT, USA).

### 2.10. ROS Measurement

PANC-1 cells were seeded at a density of 1.5 × 10^5^ cells/well in 6-well plates and cultured overnight in a 5% CO_2_ incubator at 37 °C. Then, cells were treated with eugenol for 24 h at the indicated concentrations. ROS levels were measured using dichloro-dihydro-fluorescein diacetate (DCFH-DA) and dihydroethidium. Cells were incubated for 30 min with 5 μM DCFH-DA, 30 min with 10 μM DHE, and then washed with PBS. The fluorescence intensity was measured using a CytoFLEX flow cytometer (Beckman Coulter, Brea, CA, USA) and a Nikon ECLIPSE Ts2R-FL inverted fluorescence microscope (Nikon Instruments, Tokyo, Japan).

### 2.11. Transfection (siRNA)

The cells were transiently transfected in 2 mL of OPTI-MEM (GIBCO, Carlsbad, CA, USA) containing 9 μL of Lipofectamine 3000 (Invitrogen, Carlsbad, CA, USA) with 30 nM CHOP-siRNA. After 6 h of transfection, the medium was changed to complete growth medium for 24 h. Then, cells were treated with either 400 µM eugenol, 100 ng/mL TRAIL, or both.

### 2.12. Statistical Analysis

The GraphPad Prism software (version 5.01) was used for all statistical analyses (GraphPad Software, Inc., La Jolla, CA, USA). One-way ANOVA and Tukey’s post hoc tests were used to compare groups, and an unpaired t-test was used to determine the significance between the two groups. A *p*-value < 0.05 was considered statistically significant.

## 3. Results

### 3.1. Effects of TRAIL and Eugenol on Human Pancreatic Cancer Cell Lines

Inhibition of cell proliferation by eugenol (Figure 1a) and TRAIL was analyzed in normal human pancreatic cells (HPDE) and pancreatic cancer cell lines using the WST-8 assay (Figure 1b,c). Cell proliferation in pancreatic cancer cells was inhibited in a concentration-dependent manner when treated with 200, 400, and 800 μM eugenol and cultured for 24 h (Figure 1b). As shown in Figure 1c, the pancreatic cancer cell lines were resistant to TRAIL, with the PANC-1 cell line being the most resistant among the cancer cell lines. PANC-1 cells were treated with eugenol alone, and their proliferation and regeneration abilities were visually confirmed using crystal violet and colony formation analyses (Figure 1d,e). Treatment with eugenol alone inhibited cell proliferation and cell regeneration in a concentration-dependent manner.

### 3.2. Eugenol at Subtoxic Concentrations Enhances TRAIL-Induced Cell Death in Pancreatic Cancer Cells

To evaluate whether eugenol enhances TRAIL-induced cell proliferation inhibition in HPDE and PANC-1 cells, they were co-treated with 400 μM eugenol and 100 ng/mL TRAIL for 24 h and analyzed using the WST-8 assay (Figure 2a,b). The combination of eugenol and TRAIL significantly inhibited cell proliferation in PANC-1 cells (Figure 2b); however, HPDE cells were not affected (Figure 2a). These results indicate that subtoxic concentrations of eugenol selectively sensitized human pancreatic cancer cells to TRAIL-induced inhibition of cell proliferation. Treatment of PANC-1 cells with eugenol and TRAIL, individually or in combination, and crystal violet and colony formation analyses were performed to visually confirm cell proliferation and regeneration abilities (Figure 2c,d). The proliferation and colony-forming abilities of cells were reduced by co-treatment with eugenol and TRAIL compared with those of the control cells or cells treated with each reagent alone. A wound healing analysis was performed to indirectly confirm the inhibition of cancer cell metastasis and migration by co-treatment with eugenol and TRAIL [24]. After treatment with eugenol and TRAIL individually or in combination, real-time imaging was performed every 2 h, and each image was compared 24 h later. This confirmed that the migration ability of cancer cells was inhibited in the co-treated group compared with that in the control. Therefore, the combination treatment effectively inhibited cell metastasis and migration (Figure 2e).

### 3.3. Co-Treatment with Eugenol and TRAIL Induces Apoptosis in Pancreatic Cancer Cells

PANC-1 cells were treated with 200, 400, or 800 μM eugenol and cultured for 24 h, and apoptosis was induced in a concentration-dependent manner (Figure 3a). When PANC-1 cells were treated with 10, 50, or 100 ng/mL TRAIL and cultured for 24 h, apoptosis occurred in a concentration-dependent manner (Figure 3b). To determine whether the cell death resulting from the combination of eugenol and TRAIL was due to apoptosis, we analyzed apoptosis using flow cytometry after Annexin V/PI staining. Co-treatment with eugenol and TRAIL significantly increased the apoptosis of PANC-1 cells (Figure 3c). Next, we confirmed whether PARP, caspase-8, and caspase-3 were activated in PANC-1 cells after co-treatment with eugenol and TRAIL using Western blotting. We found that activation of these proteins was increased (Figure 3d). As shown in Figure 3e, the effect of co-treatment with eugenol and TRAIL was also confirmed in the activation assay of caspase-3/7. These results indicate that subtoxic concentrations of eugenol enhanced TRAIL-induced apoptosis in pancreatic cancer cells through caspase activation.

### 3.4. Eugenol Increases TRAIL Sensitivity by Upregulating DR5

To further determine the mechanism underlying TRAIL-induced apoptosis through eugenol, apoptosis-related proteins and death receptors were identified. As shown in Figure 4a, 400 μM eugenol reduced FLIP L/S expression in PANC-1 cells. Figure 4b,c shows the Western blotting results of treating HPDE and PANC-1 cells with eugenol, which confirmed the expression of DR5 and DR4. In HPDE cells, at 400 μM eugenol, the expression of DR5 did not change and the expression of DR4 decreased, whereas, in PANC-1 cells, the expression of DR5 was significantly increased in a concentration-dependent manner but the expression of DR4 was not significantly increased. Next, we used flow cytometry to determine whether eugenol induced DR5 and DR4 expression in PANC-1 cells (Figure 4d,e). As shown in Figure 4d,e, eugenol, at a concentration of 400 μM, increased the expression of DR5, but not DR4, on the cell surface. We investigated whether the induction of DR5 expression by eugenol was specific to PANC-1 cells or if it occurred in other cell types. Figure 4f shows that eugenol inhibited pancreatic (MIA PaCa-2 and BxPC-3), breast (MDA-MB-231 and MCF-7), brain (LN-229 and U-251), and colon cancer cells (HT-29 and HCT116), indicating that eugenol induced DR5 expression. This suggests that eugenol-induced DR5 expression is not cell type specific.

### 3.5. Eugenol Induces ROS and Activates the ER Stress Pathway

To confirm whether the ER stress pathway is activated in PANC-1 cells, the cells were treated with eugenol at concentrations of 200, 400, and 800 μM for 24 h, and the expression levels of ER stress-related proteins were confirmed through Western blotting. Eugenol induced CHOP expression through the PERK pathway and increased the expression of DR5 (Figure 5a). We examined the fluorescence intensity using flow cytometry after staining with dichlorofluorescein diacetate (DCFH-DA) dye to determine whether eugenol induces the production of ROS, which causes ER stress. As shown in Figure 5b, time-course experiments showed that ROS levels increased significantly from 12 h onward and peaked at 24 h. To measure ROS in living cells, we set the eugenol treatment time to 12 h, as this time point is before the end of apoptosis, not 24 h. ROS induction was confirmed in PANC-1 cells after treatment with 400 μM eugenol for 12 h. However, it did not affect ROS production in HPDE cells (Figure 5c). Therefore, eugenol increased ROS levels in cancer cells and activated the ER stress pathway.

### 3.6. Enhanced Eugenol-Induced Apoptosis by DR5 and CHOP Upregulation Is ROS-Dependent

To demonstrate that ROS generation is involved in CHOP-induced DR5 expression, N-acetyl-cysteine (NAC), a free radical scavenger, was used [25]. In addition, the maximum expression of ROS in PANC-1 cells was confirmed using H_2_O_2_, a ROS-positive substance. Figure 6a shows that 1 h pretreatment with NAC reduced the eugenol-induced increase in ROS. In addition, ROS in the cytoplasm were measured using DCFH-DA dye and ROS in the mitochondria were measured using DHE dye in PANC-1 cells and confirmed using a fluorescence microscope. We visually confirmed that NAC pretreatment reduced the fluorescence intensity of eugenol-induced ROS (Figure 6b). We then determined whether ROS regulated eugenol-induced DR5 and CHOP expression in the presence or absence of NAC and found that NAC pretreatment of PANC-1 cells reduced the eugenol-induced upregulation of DR5 and CHOP expression (Figure 6c). Eugenol significantly increased DR5 and CHOP expression in PANC-1 cells when combined with TRAIL, whereas NAC pretreatment significantly decreased DR5 and CHOP expression (Figure 6d). As shown in Figure 6e, the eugenol/TRAIL-induced PARP cleavage significantly increased in PANC-1 cells the and NAC pretreatment significantly decreased eugenol/TRAIL-induced PARP cleavage. Additionally, as shown in Figure 6f, co-treatment with 400 μM eugenol and 100 ng/mL TRAIL significantly enhanced apoptosis, but NAC pretreatment significantly attenuated apoptosis. This suggests that ROS plays an important role in mediating the effect of eugenol on TRAIL-induced apoptosis.

### 3.7. Effects of Eugenol on Anti-Apoptotic Proteins and DR Levels of Two Human Pancreatic Cancer Cell Lines

Recent studies have shown that CHOP induced by the ER stress pathway increases DR5 expression [5]. We investigated whether CHOP played aa role in the eugenol-induced upregulation of DR5. Figure 7a shows that eugenol increased CHOP and DR5 expression simultaneously, demonstrating the relationship between DR5 and CHOP. To determine the role of CHOP in the eugenol-induced upregulation of DR5, gene silencing by CHOP-siRNA was conducted for 6 h (Figure 7b). Western blotting was performed to determine whether the presence or absence of CHOP siRNA regulated the eugenol-induced cleavage of PARP, DR5, and CHOP. We found that transfection with CHOP-siRNA significantly inhibited eugenol-induced PARP cleavage and DR5 upregulation. Next, flow cytometry was used to investigate whether CHOP inhibition by CHOP-siRNA attenuated the sensitizing effect of eugenol on TRAIL-induced apoptosis. Transfection with CHOP-siRNA significantly reduced the effects of eugenol and TRAIL-induced apoptosis (Figure 7c). These findings suggest that CHOP plays a role in the upregulation of DR5 and contributes to the sensitization of cancer cells by eugenol on TRAIL-induced apoptosis.

## 4. Discussion

TRAIL, a cytokine secreted by various immune cells, induces cancer cell death without harming normal cells [16,17,26]. However, several cancer cells lines are resistant to TRAIL. The main cause of the TRAIL resistance mechanism is the inhibition of DR4 or DR5 due to the overexpression of DcR1/2 and anti-apoptotic proteins in cancer cell lines, which inhibits the action of pro-apoptotic proteins in cells [27]. Therefore, novel sensitizers to increase the susceptibility of cancer cells to TRAIL are required.

Eugenol, a natural substance, is the main component of clove oil extracted from dried clove flower buds and steam distillation, and it has been used as a spice, analgesic, and disinfectant. Recent studies have also reported the anticancer effects of eugenol against colon and lung cancers. However, no studies have investigated whether eugenol can increase the sensitivity of cancer cells to TRAIL by overexpressing DRs. Therefore, we investigated whether pancreatic cancer cells could be sensitized to TRAIL if administered in combination with eugenol.

First, we confirmed that the combination of eugenol and TRAIL effectively inhibited the cell proliferation of PANC-1 cells but did not affect HPDE cells (Figure 2). In addition, flow cytometry confirmed that the combination of eugenol and TRAIL effectively induced apoptosis in PANC-1 cells, and increased expression of the apoptosis-related proteins PARP and caspase-8/3 was confirmed by Western blotting (Figure 3). To determine the mechanism by which eugenol increases TRAIL sensitivity in PANC-1 cells, the expression of anti/pro-apoptotic proteins was examined, and it was confirmed that eugenol inhibited FLIP, an anti-apoptotic protein. Additionally, we identified the TRAIL receptors DR4 and DR5 and found that TRAIL-induced apoptosis was enhanced by increasing the expression of DR5 (Figure 4). Eugenol did not induce the expression of DR5 in HPDE cells but decreased the expression of DR4. Several studies have shown that DR5 is upregulated through increased expression of CHOP when ER stress is activated by ROS [8,28]. In addition, p53, a tumor suppressor protein induces the accumulation of ROS in mitochondria and increases ROS levels in cells, thereby strengthening ER stress [11]. After treating PANC-1 cells with eugenol and confirming the presence of ER stress-related proteins, the PERK-CHOP pathway (p-PERK, BiP, p53, ATF-4, p-eif2α, and CHOP) was activated (Figure 5) [29,30]. In addition, staining with DCFH-DA dye to confirm whether the excessive production of ROS, which causes ER stress, was induced by eugenol, showed that ROS production increased in a time-dependent manner (Figure 5). The amount of ROS produced by eugenol in HPDE and PANC-1 cells was compared, and the results showed that ROS levels excessively increased in PANC-1 cells but not in HPDE cells. To prove that the increase in DR5 and CHOP expression was related to ROS production, cells were pretreated with NAC (N-acetyl-cysteine), an active oxygen scavenger, and then treated with eugenol. The expression of DR5 and CHOP was suppressed after NAC pretreatment, and flow cytometry and fluorescence images confirmed that ROS production was reduced by NAC (Figure 6). In addition, PARP expression and apoptosis were confirmed through flow cytometry after treatment with NAC, confirming that NAC reduced apoptosis. Finally, to demonstrate the increase in DR5 expression due to CHOP activation, CHOP siRNA was transfected into PANC-1 cells to knockdown CHOP, and DR5 expression was effectively reduced. In addition, when treated with CHOP siRNA, PARP expression was decreased, and apoptosis, as confirmed by flow cytometry, was decreased as well (Figure 7). 

One limitation of this study is the absence of animal model experiments, which indicates that the in vivo efficacy and safety of the eugenol and TRAIL combination were not evaluated. Future research should involve animal models to better ascertain and validate the therapeutic potential and safety of this combination treatment.

## 5. Conclusions

Our results showed that eugenol increased TRAIL sensitivity by upregulating DR5 through the ROS-mediated ER stress–CHOP pathway without affecting normal cells. In addition, co-treatment with eugenol and TRAIL strengthened ER stress by inducing p53 and upregulated TRAIL-induced apoptosis by inhibiting the expression of FLIP. The combination treatment of eugenol and TRAIL enhances pancreatic cancer cell death, offering a promising new approach for future pancreatic cancer therapies.

## Figures and Tables

**Figure 1 cancers-16-03092-f001:**
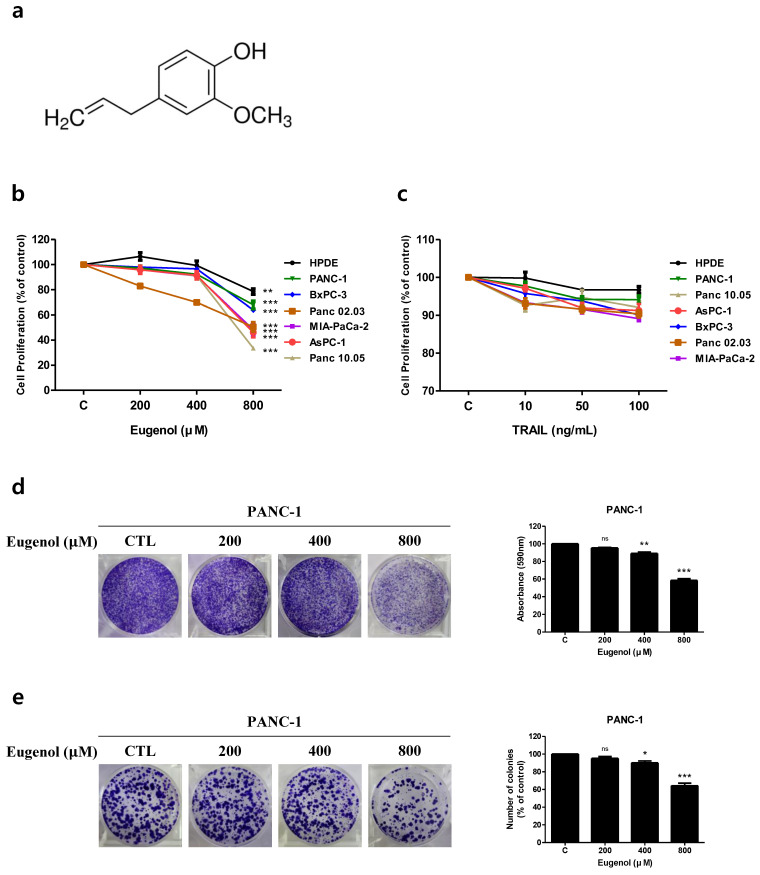
Cell proliferation inhibition assay of eugenol and tumor necrosis factor-related apoptosis-inducing ligand (TRAIL) on human pancreatic cancer cell lines. (**a**) Chemical structure of eugenol. Graphs showing the proliferation rate of various cell lines after exposure to (**b**) eugenol and (**c**) TRAIL. (**d**) In the crystal violet assay and (**e**) colony formation assay, PANC-1 cells were treated with eugenol and stained with crystal violet. Each result is presented as the mean of three independent experiments; asterisks indicate significant differences at * *p* < 0.05; ** *p* < 0.01; *** *p* < 0.001; ns = not significant.

**Figure 2 cancers-16-03092-f002:**
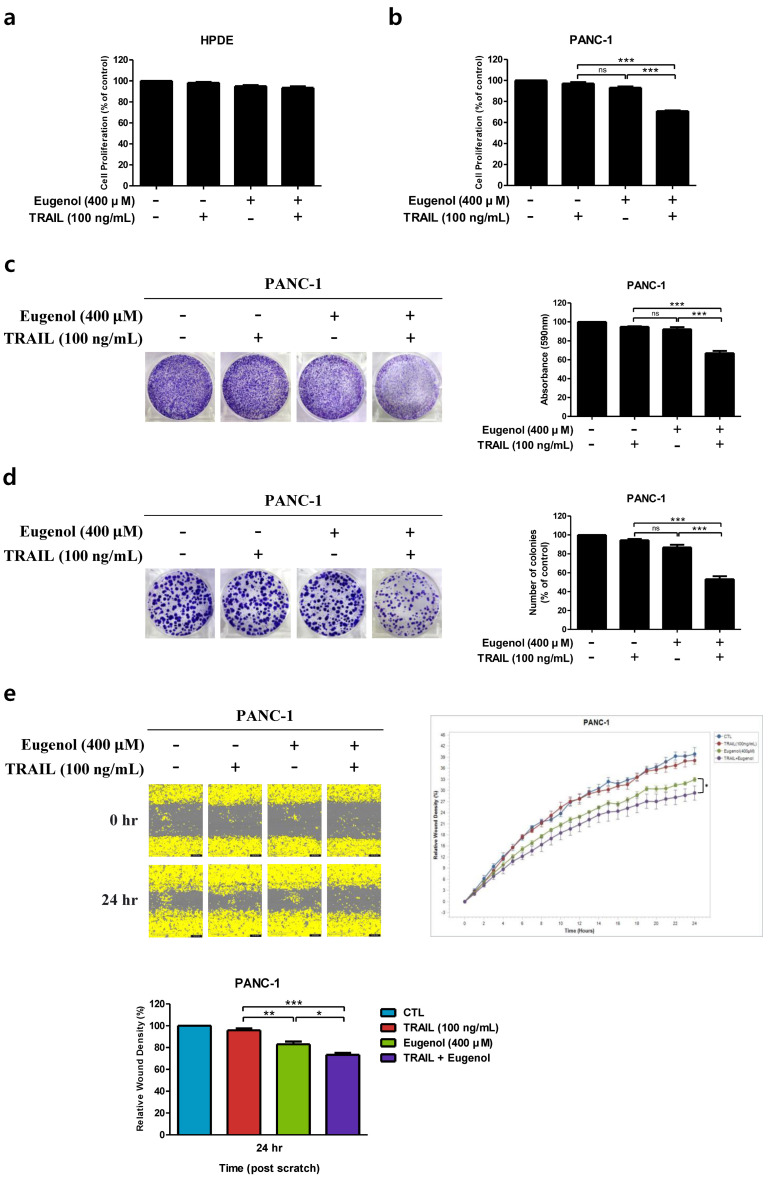
Effect of combined eugenol and TRAIL treatment on cell proliferation of PANC-1 cells. (**a**) HPDE cells and (**b**) PANC-1 cells were treated with eugenol and TRAIL; cell viability was determined. (**c**) In the crystal violet assay and (**d**) colony formation assay, PANC-1 cells were treated with eugenol and TRAIL, stained with crystal violet. (**e**) PANC-1 cells were treated with eugenol and TRAIL, and the migration of cells was assessed using a wound healing assay. Each result is presented as the mean of three independent experiments; asterisks indicate significant differences at * *p* < 0.05; ** *p* < 0.01; *** *p* < 0.001; ns = not significant.

**Figure 3 cancers-16-03092-f003:**
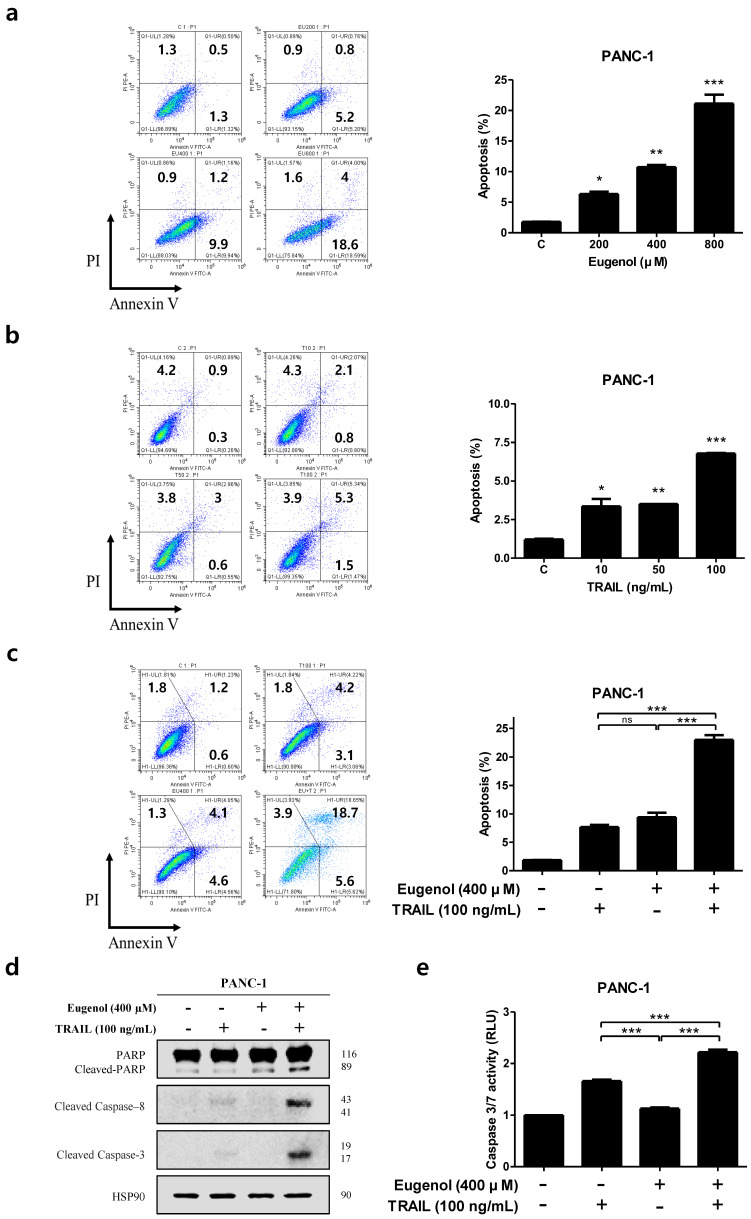
Effect of co-treatment with eugenol and TRAIL on TRAIL-induced apoptosis of PANC-1 cells. Dual fluorescence scatter plot and graphs showing the apoptosis rate of PANC-1 cells after exposure to (**a**) eugenol and (**b**) TRAIL. (**c**) PANC-1 cells treated with both eugenol and TRAIL for 24 h. Flow cytometry was performed using double staining with Annexin V and PI. (**d**) Effect of combined treatment with eugenol and TRAIL on levels of apoptosis-associated proteins. (**e**) ELISA-based luminescence assays were used to measure caspase 3/7 activities following treatment with eugenol and TRAIL. * *p* < 0.05; ** *p* < 0.01; *** *p* < 0.001; ns = not significant. The original Western blot figures can be found in Appendix A.

**Figure 4 cancers-16-03092-f004:**
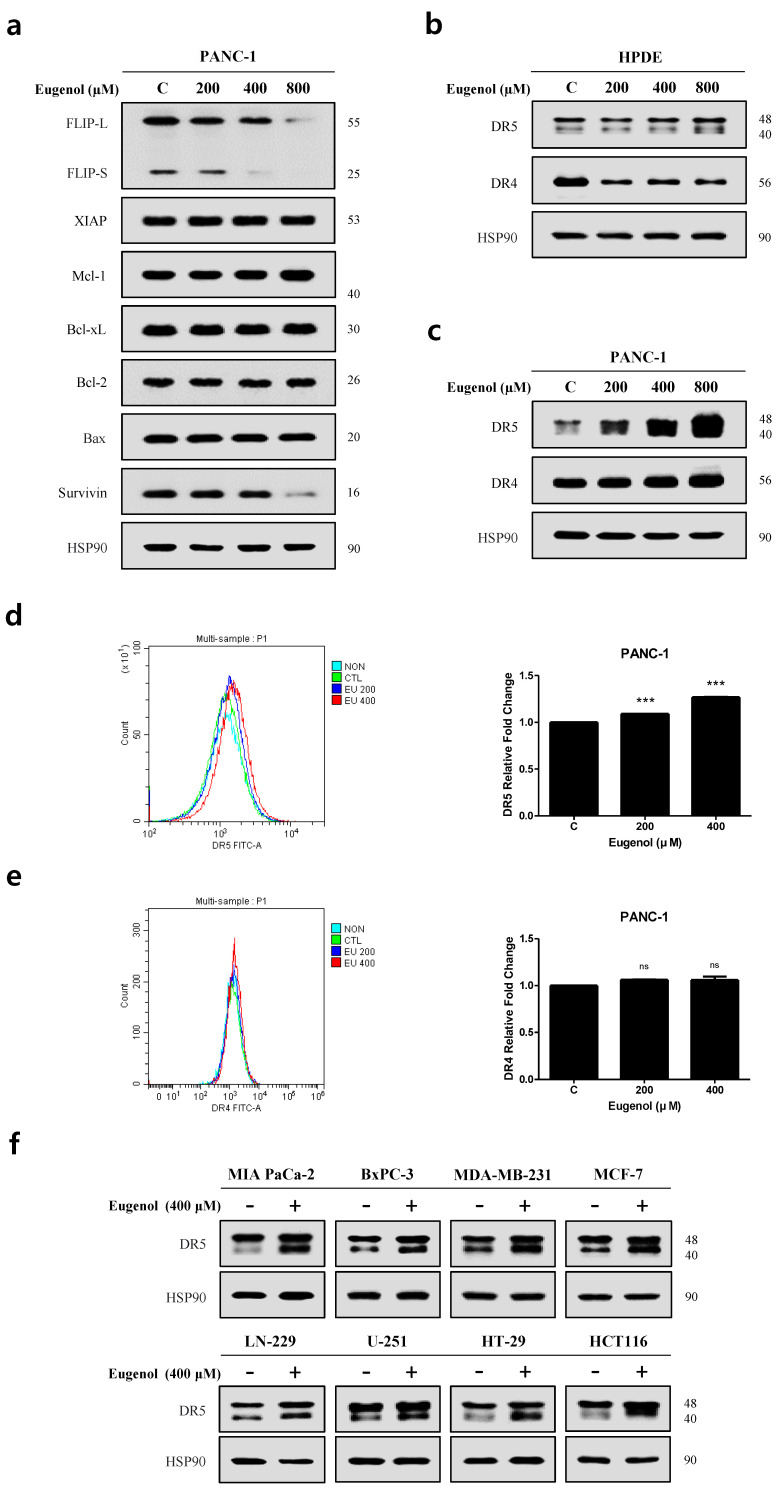
Eugenol induces DR5 expression in PANC-1 cells. (**a**) The protein levels of pro-apoptotic and anti-apoptotic proteins were examined using Western blot analysis. Death receptor expression levels after eugenol treatment in (**b**) HPDE cells and (**c**) PANC-1 cells. Flow cytometry results showing (**d**) DR5 and (**e**) DR4 expression on the cell surface. (**f**) Western blotting results showing DR5 expression from various cell lines. *** *p* < 0.001; ns = not significant. The original Western blot figures can be found in Appendix A.

**Figure 5 cancers-16-03092-f005:**
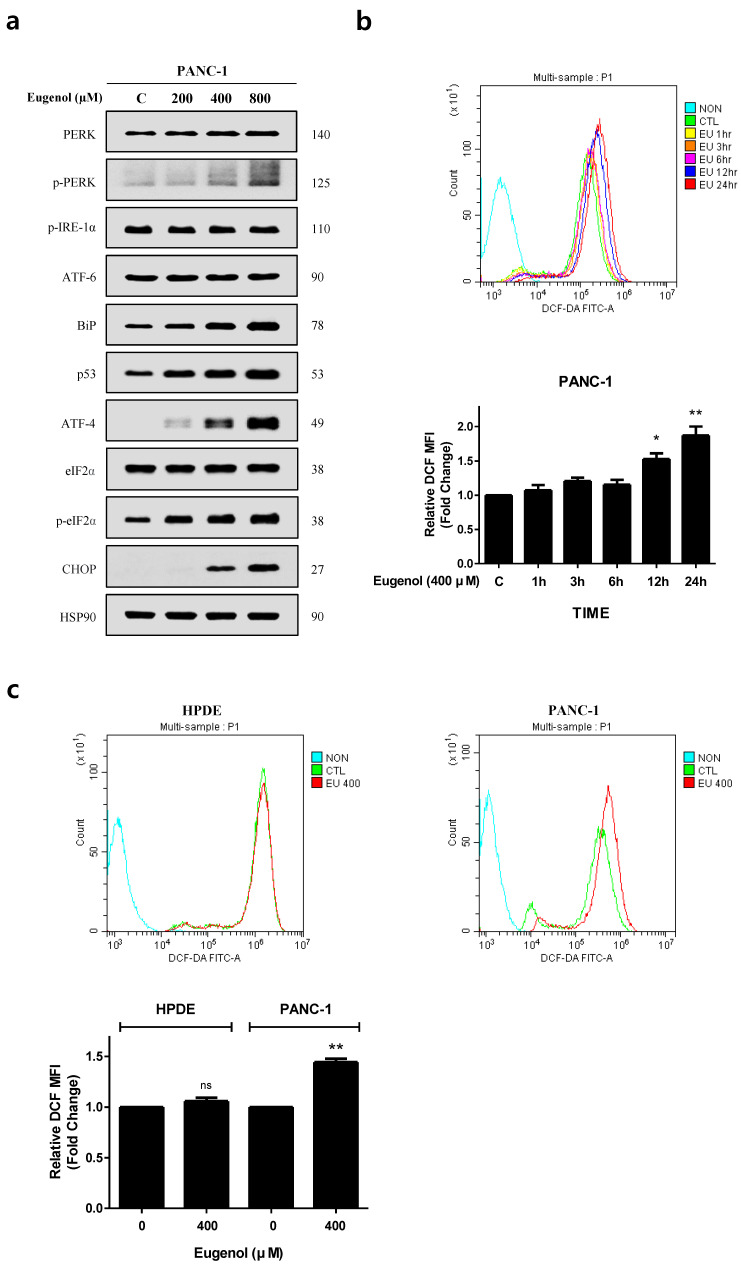
Eugenol induces ER stress in PANC-1 cells. (**a**) PANC-1 cells exposed to varying concentrations of eugenol for 24 h. The expression levels of ER stress-related proteins were analyzed using Western blotting; (**b**) PANC-1 cells treated with 400 μM eugenol and stained with DCFH-DA. (**c**) ROS production levels from the DCFH-DA analysis following 12 h treatment with 400 μM of eugenol in HPDE and PANC-1 cells. * *p* < 0.05; ** *p* < 0.01; ns = not significant. The original Western blot figures can be found in Appendix A.

**Figure 6 cancers-16-03092-f006:**
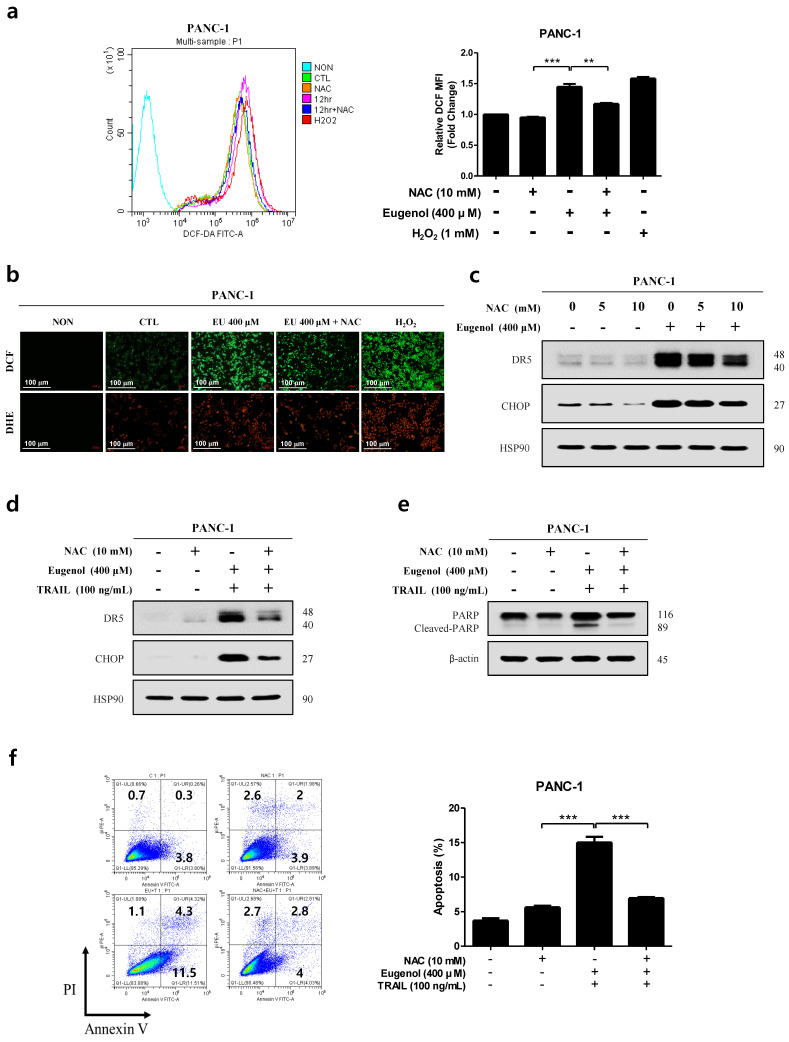
Eugenol-induced upregulation of DR5 and CHOP mediated by ROS. (**a**) PANC-1 cells pretreated with 10 mM NAC for 1 h, treated with 400 μM eugenol for 12 h, and stained with DCFH-DA. The fluorescence intensity of DCF-DA expressed in the cells was measured using flow cytometry. (**b**) PANC-1 cells pretreated with NAC for 1 h, treated with eugenol for 12 h, and stained with DCFH-DA and DHE. The fluorescence intensity of DCF-DA in the cells was determined using a fluorescence microscope at a magnification of 10×. (**c**) PANC-1 cells pretreated with NAC for 1 h and subsequently treated with eugenol for 24 h. DR5 and CHOP expression were analyzed by Western blotting. PANC-1 cells pretreated with NAC and subsequently treated with eugenol or TRAIL, followed by analysis for (**d**) DR5 and CHOP expression and (**e**) PARP and cleaved-PARP expression using Western blotting. (**f**) The dual fluorescence scatter plot and graphs illustrate the apoptosis rate of PANC-1 cells following exposure to eugenol, TRAIL, and NAC. PANC-1 cells pretreated with NAC and subsequently treated with eugenol or TRAIL were analyzed for apoptosis using flow cytometry following Annexin V and PI staining. ** *p* < 0.01; *** *p* < 0.001. The original Western blot figures can be found in Appendix A.

**Figure 7 cancers-16-03092-f007:**
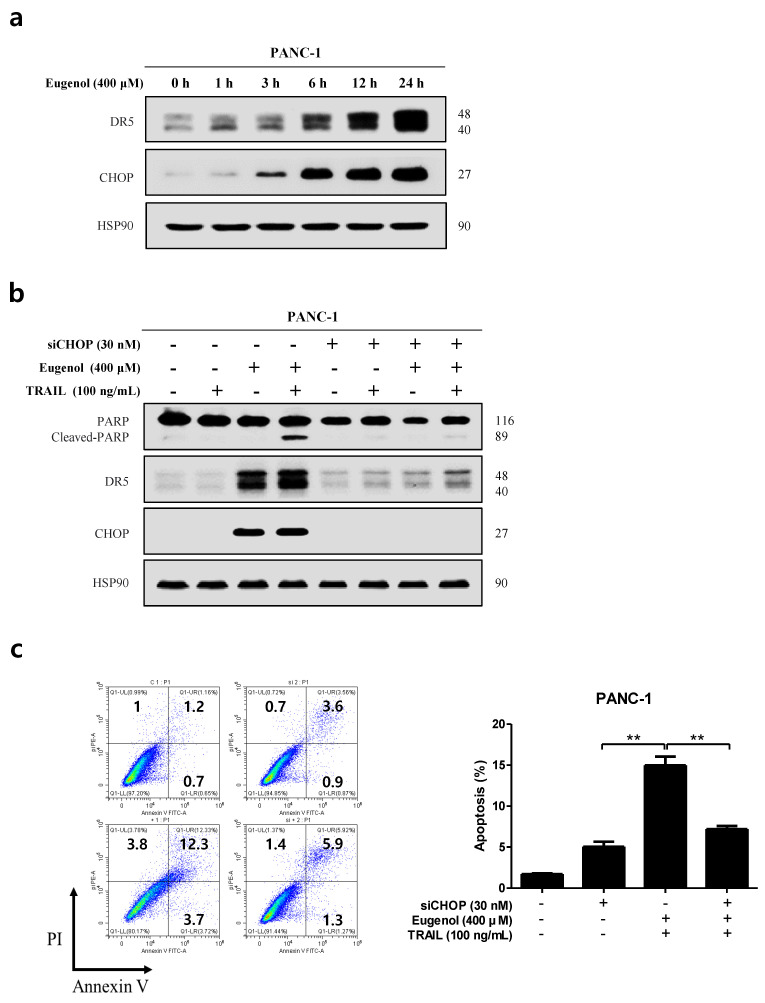
Induction of DR5 by eugenol mediated through CHOP activation. (**a**) The expression levels of DR5 and CHOP were analyzed by Western blotting in PANC-1 cells treated with eugenol over a time course. (**b**) PANC-1 cells were transfected with CHOP siRNA for 6 h and then treated with eugenol or TRAIL, followed by the determination of protein expression levels of PARP, DR5, and CHOP using Western blotting. (**c**) After transfection with CHOP siRNA, PANC-1 cells were treated with eugenol or TRAIL and subsequently stained with Annexin V and PI, followed by apoptosis analysis using flow cytometry. (**d**) Diagram of the mechanisms of action of eugenol. Pathway activation, signal propagation, or expression promotion is indicated by the symbol “⟶”, whereas pathway inactivation, signaling suppression, or expression downregulation is represented by the symbol “⊣”. ** *p* < 0.01. The original Western blot figures can be found in Appendix A.

## Data Availability

The data that support the findings of this study are available from the authors upon reasonable request.

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
