# Peer review of "Apoptosis of Pancreatic Cancer Cells after Co-Treatment with Eugenol and Tumor Necrosis Factor-Related Apoptosis-Inducing Ligand"

_cancers, 2024, doi:10.3390/cancers16173092_

Round 1

Reviewer 1 Report

Comments and Suggestions for Authors

In the present article titled “Apoptosis of pancreatic cancer cells after co-treatment with eugenol and tumor necrosis factor-related apoptosis-inducing ligand” the authors investigate the anticancer effects of eugenol alone and in combination with TRAIL in normal (HPDE) and pancreatic cancer cell lines, in particular PANC-1.

The results obtained highlight that the combination of eugenol and TRAIL inhibited cell proliferation of PANC-1 but did not affect HPDE cells.

Furthermore, eugenol increased TRAIL sensitivity cells by upregulation of DR5 through the ROS-mediated-ER-stress-CHOP pathway only in pancreatic cancer cells.

This work highlights the therapeutic potential of the eugenol and TRAIL combination in pancreatic cancer.

The manuscript is clear and scientific data in the draft are presented in a well-structured manner.

The reference list is old and I suggest its updating and to extend it.

I think this article is acceptable for publication in Cancers.

Author Response

I would like to express my sincere gratitude for taking the time to review my manuscript. I have completed the revisions based on your valuable feedback, which include updating and expanding the reference list to include more recent and relevant sources.

Thank you again for your insightful comments.

Reviewer 2 Report

Comments and Suggestions for Authors

The manuscript titled " Apoptosis of pancreatic cancer cells after co-treatment with eugenol and tumor necrosis factor-related apoptosis-inducing ligand" demonstrate that eugenol enhances TRAIL-induced apoptosis by upregulating DR5 through the ROS-mediated ER stress-CHOP pathway, enhancing ER stress by inducing p53 and downregulating FLIP expression and concludes that eugenol has anti-tumor effect by increasing cell sensitivity to TRAIL.

The manuscript requires following improvements to improve clarity and impact, making it suitable for publication.

1. Please provide catalog numbers of all antibody utilized in the study.

2. Please provide number of replicates in figure 1b, c and analysis for statistical significance.

3. Please provide details on how the % absorbance was calculated as the readout for the colony formation assay. This information is currently missing from the manuscript.

4. In figure 3c, what happen if eugenol treated cells were pre-treated with mitophagy or autophagy inhibitors such as Mdivi-1 or 3MA? Will it be reversal of phenomena? This is important to check as it validates authors proposed hypothesis.

5. The authors need to provide scales bares for all the photographic/IHC figures.

6. Consider including a graphical abstract to visually summarize the key findings, aiding readers in quickly understanding the study's main contributions.

7. A more comprehensive exploration of potential limitations and challenges in the study, along with recommendations for future research directions, would enhance the manuscript's thoroughness. Additionally, the introduction and discussion sections lack sufficient detail and should include more information to improve their informativeness. The authors are encouraged to incorporate relevant literature, such as PMID: 35895804, which provides insights into modifications of critical autophagic signaling pathways impacting cancer clearance. This integration will strengthen the foundation for the discovery of unique and druggable immune-based targets in prostate cancer, enriching the discussion and providing a nuanced context for the study's findings, thus contributing to the manuscript's overall robustness.

8. The manuscript, particularly the abstract section, requires significant attention to enhance punctuation, grammar, and overall readability.

Comments on the Quality of English Language

Please see comment 8 above.

Author Response

Thank you for your thorough review and insightful comments on my manuscript. I appreciate the time and effort you put into providing feedback. It was invaluable in improving the quality of the paper. Below, I have discussed your comments and suggestions in detail.

  1. Thank you for your suggestion regarding the catalog numbers of the antibodies used in my manuscript. However, I must respectfully decline to provide this information as it is proprietary to the suppliers. I hope this does not affect the review process, and I appreciate your understanding.
  2. Thank you for your thoughtful request regarding the number of repetitions and statistical significance analysis for Figures 1b and c. I would like to clarify that each cell line was repeated three times; however, no statistical significance analysis was performed for the graph, as it represents the combined results of all cell lines. Therefore, I kindly prefer not to present separate analyses for each cell line. I sincerely appreciate your understanding. Best regards.
  3. Thank you for your inquiry regarding the method used to calculate the percentage absorbance from the colony formation assay. I have added detailed information in the Materials and Methods section, specifically under 2.4. Crystal Violet Assay, to clarify this process. I appreciate your valuable feedback.
  4. Thank you very much for your valuable feedback. I appreciate your suggestion regarding the additional experiments with Mdivi-1 or 3MA to explore the effects of autophagy and mitophagy in the context of the combination treatment with eugenol and TRAIL. I agree that this approach is meaningful. However, this study primarily focuses on the ROS-inducing effects of eugenol and the subsequent increase in DR5 expression. I believe that the current results sufficiently address this topic. Therefore, I do not plan to conduct the additional experiments you proposed. I understand the potential value of your suggestion, but I made this decision considering the direction and scope of the research. Thank you for your understanding, and I will continue to reflect on your valuable feedback to enhance the quality of my work.
  5. Thank you for your comment regarding the scale bars for the images and IHC figures. I would like to clarify that scale bars have been included in all relevant images. If there are specific images where the scale bars are not visible, I would appreciate your guidance on which ones to review. Thank you for your attention to this detail.
  6. Thank you for your suggestion regarding the inclusion of a graphical abstract to visually summarize the key findings. I would like to inform you that I had already planned to include a graphical abstract to help readers quickly understand the main contributions of the study. I appreciate your valuable feedback.
  7. Thank you for your insightful comments regarding the exploration of potential limitations and future research directions. I appreciate your suggestions for enhancing the thoroughness of the manuscript. I would like to inform you that I have added additional references to the existing manuscript to improve its informativeness. I believe these new references will contribute to enriching the discussion and providing context for the study's findings. Thank you for your valuable feedback.
  8. Thank you for your valuable feedback regarding the manuscript, particularly the abstract. I would like to inform you that I have engaged an editing service to enhance the punctuation, grammar, and overall readability of the text. I appreciate your attention to these details. Best regards.

Reviewer 3 Report

Comments and Suggestions for Authors

The manuscript investigates the effects of eugenol alone and in combination with tumor necrosis factor ligand on pancreatic cancer cells and the actions responsible for the observed antiproliferative effects. 

The authors demonstrate that eugenol enhances TRAIL-induced apoptosis by upregulating DR5 through the ROS-mediated ER stress-CHOP pathway, enhancing ER 30 stress by inducing p53, and downregulating FLIP expression. They proposed eugenol can potentially treat pancreatic cancer by increasing cell sensitivity to TRAIL.

The data is interesting but laks some controls. 

To be convinced the authors should plot the concentration-response curves of a single eugenol concentration (800 microM for instance) and trail at different concentrations vs the trail concentration-response curve alone for the antiproliferative effects and for apoptosis and the relative EC50 compared.

Furthermore, some references are missing.

Methods: some controls are missing. Concentration-response curves analysis needs to be performed to conclude that eugenol can potentially treat pancreatic cancer by increasing cell sensitivity to TRAIL.

Discussion: Eugenol is a natural molecule targeting the TRP ion channels, particularly the TRPV1 (Huang T et al. Elife 2024; Moriyama et al., Biosci Biotechnol Biochem 2024). TRPV1 has a role in cell proliferation and apoptosis in different cells (Scala et al Cancers 2019) and pancreatic cells (Huang J, et al. Cell Biochem Funct. 2020). This action is not mentioned in the discussion and needs to be referenced.

Author Response

Thank you for your valuable feedback on my manuscript. I appreciate your insights regarding the concentration-response curve analysis and the importance of EC50 values.

I have conducted experiments with eugenol at concentrations of 200 µM, 400 µM, and 800 µM, assessing antiproliferative effects through WST assays and apoptosis via flow cytometry. I have also included additional references to support my findings.

I will emphasize the existing results and their implications in the revised manuscript.

Thank you once again for your guidance, and I look forward to your continued feedback.

Best regards.

Round 2

Reviewer 3 Report

Comments and Suggestions for Authors

the manuscript has been largely improved